# Microevolution of *Neisseria lactamica* during nasopharyngeal colonisation induced by controlled human infection

Anish Pandey[1], David W. Cleary [1,2,3], Jay R. Laver[1], Andrew Gorringe[4], Alice M. Deasy[5,6], Adam P. Dale[1,2], Paul D. Morris[5,6], Xavier Didelot[7,9], Martin C.J. Maiden [8] & Robert C. Read [1,2,3]

*Neisseria lactamica* is a harmless coloniser of the infant respiratory tract, and has a mutually-excluding relationship with the pathogen *Neisseria meningitidis*. Here we report controlled human infection with genomically-defined *N. lactamica* and subsequent bacterial microevolution during 26 weeks of colonisation. We find that most mutations that occur during nasopharyngeal carriage are transient indels within repetitive tracts of putative phase-variable loci associated with host-microbe interactions (*pgl* and *lgt*) and iron acquisition (*fetA* promotor and *hpuA*). Recurrent polymorphisms occurred in genes associated with energy metabolism (*nuoN*, *rssA*) and the CRISPR-associated *cas1*. A gene encoding a large hypothetical protein was often mutated in 27% of the subjects. In volunteers who were naturally co-colonised with meningococci, recombination altered allelic identity in *N. lactamica* to resemble meningococcal alleles, including loci associated with metabolism, outer membrane proteins and immune response activators. Our results suggest that phase variable genes are often mutated during carriage-associated microevolution.

[1] Clinical and Experimental Sciences, Faculty of Medicine, University of Southampton, Southampton SO166YD, UK. [2] Southampton NIHR Biomedical Research Centre, University Hospital Southampton, Southampton SO166YD, UK. [3] Institute for Life Sciences, University of Southampton, Southampton SO166YD, UK. [4] Public Health England, Porton Down, Salisbury SP40JG, UK. [5] Department of Infection, Immunity and Cardiovascular Disease, University of Sheffield, Sheffield S103JF, UK. [6] Department of Cardiology, Sheffield Teaching Hospitals NHS Foundation Trust, Sheffield S103JF, UK. [7] School of Public Health, Faculty of Medicine, Imperial College London, London SW72AZ, UK. [8] Department of Zoology, Peter Medawar Building, University of Oxford, Oxford OX13SY, UK. [9] Present address: Department of Statistics, School of Life Sciences, Gibbet Hill Campus, University of Warwick, Coventry CV4 7AL, UK. Correspondence and requests for materials should be addressed to A.P. (email: anishkumarpandey89@gmail.com) or to R.C.R. (email: r.c.read@soton.ac.uk)

Within-host selection of some bacterial species has been investigated in a variety of human niches during natural disease[1]. In the respiratory tract, startling levels of genomic change have been observed in pathogens during disease. Examples include diversifying hypermutation among *Burkholderia dolosa* populations[2] and convergent evolution in *Pseudomonas aeruginosa* sub-lineages, both among sufferers of cystic fibrosis[3,4].

At the gene-level, mutations within a small number of loci, mediating DNA mismatch repair systems, can lead to hypermutation among host-pathogen interaction genes[2]. Genetic diversity can also be caused by phase variation resulting from random slipped-strand mispairing in hypermutable, repetitive, polymeric sequences in 'contingency loci', such as the lipooligo-saccharide biosynthetic genes[5–7]. These can drive host- or population-specific, selection bottlenecks[8]. The result of this is phenotypic and stochastic variation in both colonisation and disease outcomes, which is enhanced by the acquisition of beneficial genetic material from unrelated organisms through the various mechanisms of horizontal gene transfer[9].

Colonisation is an important prerequisite for disease[6], but beyond species-species interactions and clearance, very little is known about how colonising species are impacted by the cumulative challenges experienced when occupying a new niche. Asymptomatic carriage of *Staphylococcus aureus* in the anterior nares is associated with rare adaptive change, but invasive isolates exhibit very little genomic variation compared to carriage isolates[10,11]. Regarding the nasopharyngeal pathogen *Neisseria meningitidis*, a study contrasting bloodstream and nasal isolates from four individuals found no evidence of gene-specific convergent evolution but detected phase variation in type IV pilus biogenesis loci[12].

*Neisseria lactamica* is an exclusively human nasopharyngeal commensal with highest prevalence of carriage in infancy. Carriage of *N. lactamica* has an inverse epidemiological relationship with infections caused by *N. meningitidis*[13]. Here, we used two controlled human infection studies to examine the microevolution of *N. lactamica*, a harmless commensal, genetically and phenotypically similar to *N. meningitidis* and *Neisseria gonnorhoeae*[14]. Initially, a short-term colonisation study was used to contrast *N. lactamica* collected from in vivo and in vitro passage over a period of 1 month. A second study, the purpose of which was to explore the potential, protective effect of *N. lactamica* colonisation in young adults, a demographic at risk of *Neisseria* infection[15], examined inoculated *N. lactamica*, other wild-type *N. lactamica* and wild-type *N. meningitidis* collected[16], over 6 months of carriage.

In this work, we determine the intra-host genomic diversification of *N. lactamica* over time, comparing this with in vitro dynamics. We contrast the contribution of non-synonymous polymorphisms to that of mutations in contingency loci that drive phase variation. In addition, we examine the frequency of recombination, inferring horizontal gene transfer (HGT) events between both inoculated and wild-type *N. lactamica* with *N. meningitidis*. Using estimates of diversity, we determine the effective population size in colonised individuals. These results offer insights into the impacts on genomic structure and microevolutionary processes in a bacterial coloniser of the human respiratory tract.

## Results

**Mutations and effective population size.** We recovered isolates from throat swabs taken from volunteers who had been inoculated with $10^4$ CFU ml$^{-1}$ of *N. lactamica* Y92–1009 cell bank stock[17] for a short period (1 month) of carriage and compared

them with isolates that underwent in vitro passage for the same period of time (Supplementary Figure 1; additional data in Supplementary Data 1). Isolates that had undergone passage in vitro exhibited a greater number of non-synonymous and stop codon single nucleotide polymorphisms (SNPs). There were also more insertions ($n = 6$ base insertion events) observed during in vitro passage. Isolates from colonised participants had marginally more synonymous SNPs but no insertions, only one deletion, and no SNPs introducing premature stop codons. Deleterious mutations observed in the in vitro isolates involved citrate synthase (*gltA*) and pilin subunit (*pilP*) genes and, once fixed in the population, were seen to recur in all subsequent isolates. This contrasts with isolates recovered from volunteers who had undergone controlled infection (ie in vivo), in which almost all SNPs occurred only once per sampling point despite multiple colony sampling. Mutations in *mqo* (Malate:quinone oxidoreductase) and *phrB* which confers resistance to nalidixin and rifampicin in *N. gonorrhoeae*[18], were seen to occur in both datasets; however, the position of *mqo* mutations differed while the synonymous *phrA* mutation occurred in the same position in both the in vitro controls as well 2/5 colonised volunteers (Supplementary Data 1)

A greater number of mutations in genes containing hypermutable, polymeric tracts, i.e. phase variable genes, were observed in vivo. These were compared to those identified in vitro over a month. No unique in vitro mutations were identified (i.e. these were also seen in vivo); conversely, novel in vivo changes in repetitive sequences were observed. Two of these genes were primarily associated with the modification of lipooligosaccharide (LOS), known to mediate host-pathogen response (glycosyltrasnferases; *lgtG* gene). There was also phase variation detected in the specificity subunit (*hsdS*) of the type 1, dual gene, NgoAV restriction modification system and genes implicated in iron acquisition among *Neisseria* spp. (*hpuA* and rubredoxin). Phase variation was also observed in a conserved hypothetical protein with putative glycosyl transferase activity (Table 1). No consistent pattern of phase switching ON vs phase switching OFF was observed among the isolates tested (Supplementary Data 2), however, two genes, the haemoglobin receptor *hpuA* and glycosyl*transferase lgtG* exhibited a significant shift from PHASE ON -> OFF in weeks 8 and 4 respectively ($<p = 0.05$ Fisher's exact test).

Substitution rate estimates were five times greater in the in vitro isolates ($7.55 \times 10^{-5}$ substitutions per site per year) compared to the in vivo isolates ($1.45 \times 10^{-5}$ substitutions per site per year, extrapolated from 1 month of carriage). Tajima's D, a comparison of pairwise differences and segregating sites in a given population of alleles, was used to indicate whether the population was undergoing positive (selective sweeps) or negative selection. Here the statistic was negative (approximately $-1$) for both in vivo and in vitro (Table 2), an indication of selective sweeps and a function of rare mutations in the population. Additionally, the estimates of effective population size ($N_e$) were low (~18) and approximately the same for both sample sets. Whilst differences appeared in both the rate and type of mutation that occurred during in vivo colonisation, as compared to in vitro passage, the overall impact on population genomic diversity was similar.

**Microevolution during long-term *N. lactamica* carriage in humans.** In a controlled human infection model, in which 149 volunteers were inoculated with *N. lactamica* intranasally, a total of 95 isolates of *N. lactamica* Y92–1009 and 14 isolates of *N. meningitidis* from 37 participants recovered over 6 months[16] were sequenced. An additional two individuals carried wild-type *N. lactamica* belonging to sequence types other than the inoculum

**Table 1 Phase-variable locus mutations observed in comparison of in vitro passage and in vivo colonization during short-term carriage (1 month)**

| Position | Common genotypes[a] | In vivo only genotypes | Location[b] | Gene/(s) | Details |
|---|---|---|---|---|---|
| 1,483,216 | (G)10→9 | (G)10→11 | Coding | NEIS1156 (c-hp) | Glycosyl transferase |
| 161,341 | (C)12→13 | (C)12→11 | Coding | NEIS2011 (lgtG) glucosyltransferase | LOS subunit modifierby transferase activity |
| 1,124,158 | (G)9→10 | (TACGCTGGAAGC) 1–2 | Coding | NEIS2362(36) hsdS | Type I restriction modification system, specificity subunit S |
| 1,124,237 | (G)9→10 | (G)9→11 | Coding | NEIS2362(36) hsdS | Type I restriction modification system, specificity subunit S |
| 1,483,216 | N/A: in vivo only | (C)10→11 | Coding | glycosyl transferase family 2 | Glycosyl transferase gene |
| 617,862 | N/A: in vivo only | (C)10→9 | Coding | hpuA | Haemoglobin-haptoglobin utilisation protein |
| 1,688,136 | N/A: in vivo only | (G)13 →12 | Intergenic | l-hp/rubredoxin | Large multidomain protein[c]/Rubredoxin: involved in $Fe^{2+}$ binding |
| 1,717,988 | N/A: in vivo only | (C)10→9 | Intergenic | ybaB/MDP | DNA binding protein /multidomain protein |
| 1,933,087 | (A)28→27 | (A)28→26 | Intergenic | AutA /hp | Autotransporter A |
| 593,527 | (G)10→9 | (G)10→12 | Intergenic | FetA VRF4-8 ←/→ NEIS1949 | Finetyping antigen/GroS chaperone protein folding |
| 189,023 | (A)20→19 | (A)20→17 | Intergenic | NEIS2043: 2 (Thif) →/← NEIS2044: 28 | Enzyme activation/hp |
| 837,301 | (CTTG)9→10 | (CTTG)9→11 | Intergenic | hp ←/→ NEIS0842:42 | YadA-like C-terminal region/ conserved hypothetical protein |
| 893,705 | (T)18→17 | (T)18→15 | Intergenic | NEIS0978 (396) ←/← NEIS0528(112) | Putative surface fibrial protein/putative periplasmic binding protein |
| 893,707 | (T)18→17 | (T)18→19 | Intergenic | NEIS0978 (396) ←/← NEIS0528(112) | Putative surface fibrial protein/putative periplasmic binding protein |

[a]Common genotypes are those that were found in both in vivo and in vitro datasets
[b]Locations are wither within coding sequences or intergenic. For the latter this simply refers to a location outside a known CDS, no inference to upstream promoters has been made
[c]Multi Domain Protein contained the following domains (yadA Hia auto transporter, chromosome segregation ATPase, flagellin)

**Table 2 Effective population size and Tajima's D calculated for _N. lactamica_ Y92–1009 sample sets**

| _N. lactamica_ Y92-1009 sample set (n) | Tajima's D | θ Tajima (+/− s.d.) | Effective population size ($N_e$) |
|---|---|---|---|
| In vitro passaged (n = 29)[a] | −1.08 | 0.13 (+/−0.11) | 18.74 |
| In vivo (n = 31)[a] | −0.92 | 0.12 (+/−0.12) | 17.52 |
| In vivo _N. lactamica_ solo colonised (n = 97)[b] | −1.02 | 0.20 (+/−0.18) | 29.79 |
| In vivo _N. lactamica_ co-colonised with _N. meningitidis_ (n = 19)[b] | −0.96 | 0.26 (+/−0.23) | 39.10 |

Sample sets marked with
[a]Were sourced from a short-term clinical study in which isolates were recovered from inoculated volunteers as opposed to in vitro passage. This study took place over a month. Sample sets marked with
[b]Were sourced from a long-term, 26-week colonization, experimental human challenge study. S.d. is standard deviation

strain (ST-4192 and ST-11524, n = 9 isolates) and are discussed later. In 91% (n = 32) of participants, _N. lactamica_ was isolated from more than one sampling point over the course of 6 months. In 8 individuals both _N. lactamica_ and _N. meningitidis_ were isolated, with 4 individuals having simultaneous carriage of both species.

A total of 268 mutations were identified in _N. lactamica_ following isolation. The most prevalent (71.6%, n = 192), were modifications to repetitive tracts in putatively phase variable loci. A list of genes containing 6–14 nucleotide homopolymeric tracts in the _N. lactamica_ Y92-1009 are detailed in Supplementary Data 3. From this list of 98 genes, 11 contingency loci were found in which phase variation recurred longitudinally, within and between volunteers (Fig. 1a, b). All contingency loci were found to contain homopolymeric G/C tracts, despite evidence of A/T tracts among the 98 candidate loci. Genes affected by these mutations were involved in LOS and pilin subunit modification (_lgt_ and _pgl_ loci), iron acquisition (_hpuA_) and type I restriction-modification (_hsdS_). The hypothetical proteins were assigned putative function as glycosyl transferases (hp and c-hp), and

transferrin binding activity (hp2). While most of these mutations occurred in polymeric tracts located within coding sequence, 28/192 were detected in an intergenic promotor region bordering the _fetA_ gene, potentially modulating transcriptional efficiency. All isolates contained changes in the repeat tracts of at least two of these contingency loci.

Transient SNPs in non-phase varying genes were the second most abundant mutations (46/268, 17%). These occurred after various intervals within different volunteers and were detected once. A total of 29 non-synonymous SNPs, 15 synonymous SNPs, 3 insertions, 2 multiple base substitutions and 3 deletion events were observed (Supplementary Data 4).

We defined a persistent mutation as one which was identified in consecutive isolates. A total of 15 SNPs, disseminated among five of the volunteers, were found to persist longitudinally. These were found in lipid-hydrolase protein _rssA_, oxidoreductase _nuoN_, energy synthase _atpC_ and CRISPR-associated endonuclease _cas1_ and two hypothetical proteins. In addition to this, seven deletion events appeared fixed in three volunteers, these affected a large hypothetical protein and are discussed later in this section. The

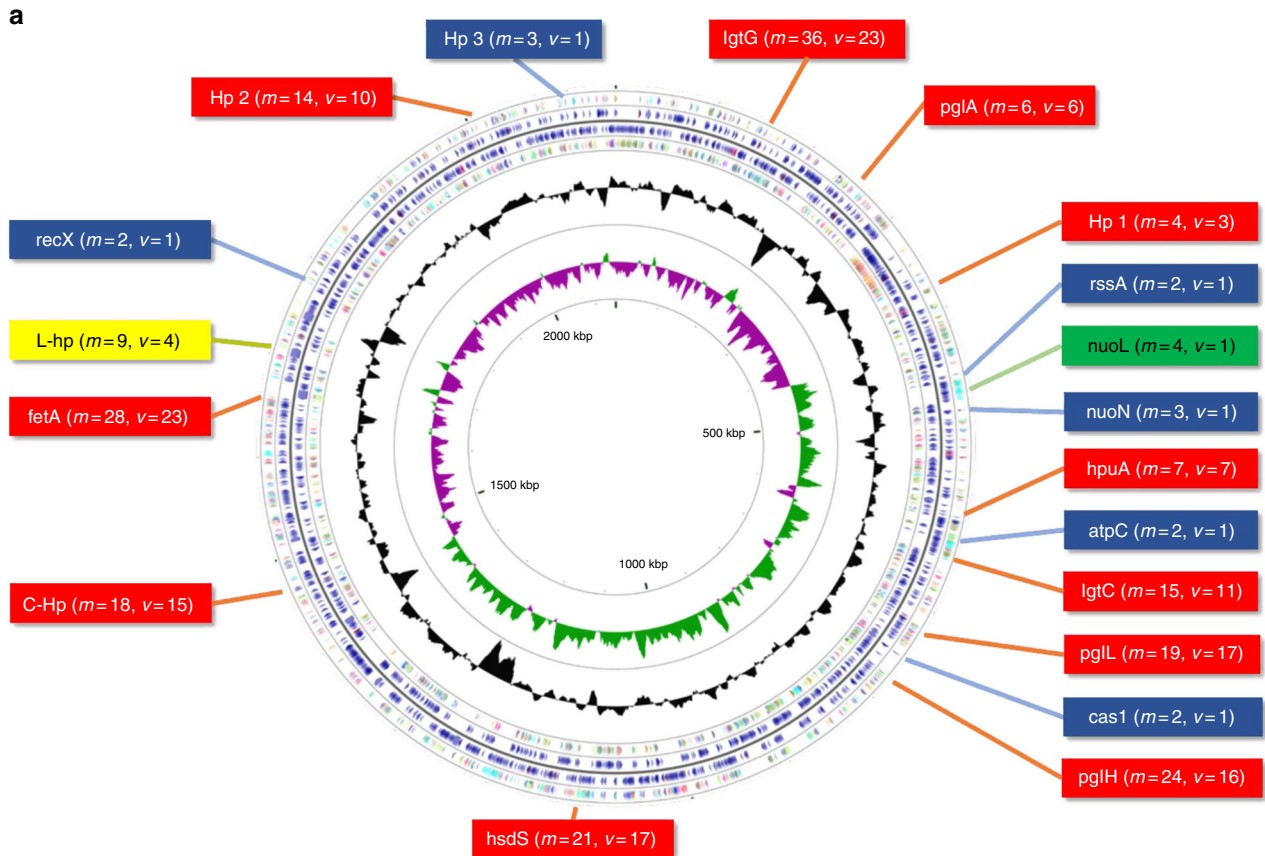

**a**

| Volunteer | Position | Mutation Type | Gene | Description |
|---|---|---|---|---|
| Multiple | 161341 | PV-C Tract | lgtG | Lipooligosaccharide glycosyl transferase G |
| Multiple | 617862 | PV-C Tract | hpuA | Hemoglobin-haptoglobin utilization protein A |
| Multiple | 1674201 | PV-C tract | fetA | Iron regulated outer membrane protein |
| Multiple | 256710 | PV-G Tract | pglA | Pilin glycosylation A |
| Multiple | 386523 | PV-G Tract | hp | Hypothetical protein (putative transferase activity) |
| Multiple | 659619 | PV-G Tract | lgtC | Lipooligosaccharide glycosyl transferase C |
| Multiple | 742750 | PV-G Tract | pglL | Pilin glycosylation L |
| Multiple | 776735 | PV-G tract | pglH | Pilin glycosylation H |
| Multiple | 1124237 | PV-G tract | hsDS | Type-1 restriction enzyme specificity protein MPN_089 |
| Multiple | 1483216 | PV-G tract | c-hp | Conserved hypothetical protein (glycosyl transferase, (NEIS1156)) |
| Multiple | 2004035 | PV-G Tract | hp2 | Hypothetical protein (putative transferrin binding activity) |
| 160 | 468817 | SNP | rssA | NTE family protein RssA |
| AN Week 2 | 500676 | SNP | nuoL | NADH-quinone oxidoreductase subunit L |
| 190 | 505463 | SNP | nuoN | NADH-quinone oxidoreductase subunit N |
| 264 | 653511 | SNP | atpC | ATP synthase epsilon chain |
| 63 | 760996 | SNP | cas1 | CRISPR-associated endonuclease Cas1 |
| 88 | 1804594 | SNP | recX | Regulatory protein RecX |
| 264 | 2123920 | SNP | hp | Roadblock/LC7 domain protein |
| Multiple | 1719272 | Various | l-hp | Large multi domain protein |

**Fig. 1 a, b** Mutations that occurred in isolates recovered over 6 months of nasopharyngeal carriage are outlined as follows: changes to repetitive tracts of phase-variable, contingency loci (Red), genes in which SNPs were persistently detected across multiple volunteers (Blue), including an example from the short-term, in vitro VS in vivo study (Green) and the sole gene in which various mutations discovered across multiple timepoints and volunteers (Yellow). $m$ refers to the number of mutations detected in a given gene across $v$ volunteers. The genes are placed in positions relative to their approximate location in the *N. lactamica* Y92-1009 genome. **b** Volunteers in which Single-Nucleotide Polymorphisms (SNPs) were seen are specified in the Volunteer column. Mutations which affected multiple volunteers are quantified in the accompanying figure above. Mutation locations are indicated in the Position column. The type of mutation is given in the mutation column where C tract/G tract refers to a homopolymeric region of cytosine/ guanine repeats where insertions and deletions occurred. PV denotes phase-variable

non-synonymous mutation found in *rssA* has been seen in another *N. lactamica* isolate found via BLASTp against the nr database. The location and nature of the remaining mutations were not identified in either sequence database blast of the entire catalogue of pubMLST Neisseria or via non-specific BLASTp against the nr database. Therefore, these are most likely novel polymorphisms found during the course of this study. These persistent mutations were shown alongside the phase variations in Fig. 1a, b and tabulated in Supplementary Data 5.

Convergent mutation was rare in this dataset; however, twelve mutations were found in eight (27%) of the volunteers that localized to a single, large (~10 kb) hypothetical protein (herein referred to as L-hp, accession: ARB05049.1) (Fig. 2). The sequence of this gene is specific to this strain and not detected outside of *N. lactamica* Y92-1009 in a sequence search among all *Neisseria* isolates in the pubMLST.org/neisseria database or non-specific BLASTp against the nr database. The locus is located at position 1,719,272 in the *N. lactamica* Y92-1009 genome, with a GC content (31.3%) that was skewed negatively, compared to the overall GC content of the genome (52.2%) closer towards the mean GC content for *Haemophilus influenzae* (~G + C: 38%). This was the only coding sequence devoid of phase variable tracts found to be targeted by mutation across multiple volunteers. L-hp contained 8 conserved protein domains which included an extended signal peptide of the type V secretion system (pfam13018), A tryptophan-ring motif (pfam15401), supporting a HiaBD2 trimeric autotransporter adhesin (pfam15403), Yad-A like, left-handed beta roll (cl17507) and four YadA-like C terminal regions (3 copies of cl27224 and pfam03895). Five of the twelve mutations persisted in volunteers 158 and 227, reducing the predicted protein size by 84–93%. Only two of the twelve mutations identified were seen to occur within the domains and both targeted YadA-like C terminal regions. CDART analysis, which contrasts the positions and functions of protein domain architecture but not necessarily sequence similarity, revealed significant homology with other *N. lactamica* strains (score = 5/5, three sequences) and *H. influenzae* (Mean GC content: ~38, score 4/5, thirteen sequences) and *Neisseria cinerea* ATCC 14685 hep/hag repeat protein (score = 4/5, two sequences) but only marginal or non-existent matches to other members of the Neisseriaceae. This indicates that three other *N. lactamica* strains possess this exact protein architecture (with regards to domain dissemination and function throughout the protein) with all five domains present, but that fifteen sequences from both *H. influenzae* and *N. cinerea* genomes possess proteins with very similar domain architecture. Protein BLAST analysis identified significant matches (>95% identity) only among other *N. lactamica* Y92-1009 genomes.

**Effect of recombination**. ClonalframeML was used to detect loci in *N. lactamica* affected by recombination in the core genome (*n* = 1595 core loci) and to quantify the total effect of recombination compared to mutation (r/m) (Table 3). A total of 18 loci were detected as recombinant over the course of this analysis (Table 4). Three of these loci affected by recombination were identified in volunteers (*n* = 27 volunteers) carrying only *N. lactamica* Y92-1009. These included a hypothetical protein (NEIS1708, NMB0444), a putative mafs3 cassette (NEIS1794) and an intra-cellular septation protein (NEIS1828, NMB0342). The effect of recombination (r/m) was 3.62, based on all volunteer isolates experimentally challenged with *N. lactamica* Y92-1009. Eight volunteers co-carried the experimentally inoculated *N. lactamica* Y92-1009 strain, however isolates from one volunteer were excluded from recombination analysis as only one isolate from *N. meningitidis* and *N. lactamica* was recovered leaving seven

volunteers (*n* = 19 isolates) with wild-type *N. meningitidis* (Table 3). Here the average r/m value was 3.84, which was similar to that seen when the bacterium was not co-colonised with *N. meningitidis*. Only small (<20 bp) importations were detected in volunteers co-carrying experimentally inoculated *N. lactamica* Y92–1009 and *N. meningitidis*. As a result, there were no recombinant loci identified in co-colonised individuals who were experimentally inoculated with *N. lactamica*.

**Mutation rates and effective population size**. Given the above mutation parameters we estimated the effective population size for the *N. lactamica* inoculum. We inferred a mutation rate in terms of SNPs per site per year. Sixty-four SNPs were detected among twenty-seven volunteers carrying only *N. lactamica* Y92–1009. This yielded a mutation rate of $2.21 \times 10^{-6}$ substitutions per site per year. Seven SNPs were detected among seven volunteers co-colonised with *N. meningitidis* and were used to infer a mutation rate of $9.32 \times 10^{-7}$ substitutions per site per year. Therefore, no evidence of an enhanced mutation rate was observed in volunteers co-carrying the inoculated *N. lactamica* and strains of *N. meningitidis*. Tajima's D was found to be negative (approximately −1) for *N. lactamica* isolates examined with or without meningococcal co-colonisation (Table 2). The estimations of effective population size ($N_e$) were found to be consistently low in all four sets (Table 2).

**Recombination between co-colonising *N. lactamica* and *N. meningitidis***. Two volunteers (36 and 291) were found to co-carry naturally acquired *N. lactamica* (ST-4192 and 11524, *n* = 9 isolates) in addition to commensal *N. meningitidis* (ST-41 and ST-10457, *n* = 6 isolates) (Supplementary Table 1). Together, these volunteers accounted for the majority of recombinant loci detected (*n* = 15/18 loci, Table 4). Volunteer 255 was the only individual who was observed to carry more than one *N. meningitis* ST during this period: a-ST-10478 meningococcus at the first sampling point, followed by a ST-213 *N. meningitidis*. In contrast to the *N. lactamica* Y92-1009 challenge strain, the r/m values were between 3–5 times higher for wild-type *N. lactamica* and *N. meningitidis* (Table 3). The largest r/m (overall effect of recombination calculated as described in Table 3 legend.) detected was 18.06 and was calculated for volunteer 291 wild-type *N. lactamica* and may imply the organism was subject to a recent, interspecific recombination event although it is impossible to verify whether the event occurred before or during the course of the study.

Six recombinant alleles were identified in the *N. lactamica* isolated from volunteer 291 that were also found in the co-carried *N. meningitidis* (Supplementary Figure 2). Of these six alleles, four (NEIS0071, a putative lipoprotein; NEIS0072 & NEIS2514, hypothetical proteins; and NEIS0899, *lst*, a LOS subunit sialyltransferase) were shared by *N. lactamica* and co-colonising *N. meningitidis* recovered from the second and third sampling points. By week 16 however a different allele for these loci was identified in the recovered *N. lactamica*. In contrast, for two loci (NEIS1834, an acyl carrier protein and NEIS1835, a succinyl-transferase) it was the week 16 isolates that were found to share the same allele as the meningococci. This might suggest that these recombination events reflect independent exchanges, however this is not possible to determine from this data alone.

**Discussion**

As anticipated from previous in vitro studies of the meningo-coccus[19], phase variation was the principle mechanism by which microbial diversity arose in the commensal carried by individuals over 6 months. With the assumption that the mutation rate estimate acquired from the largest number of volunteers (*n* = 97)

**a** Domains, mutations and mutation ranges introduced in large-hypothetical protein (ARB05049.1)

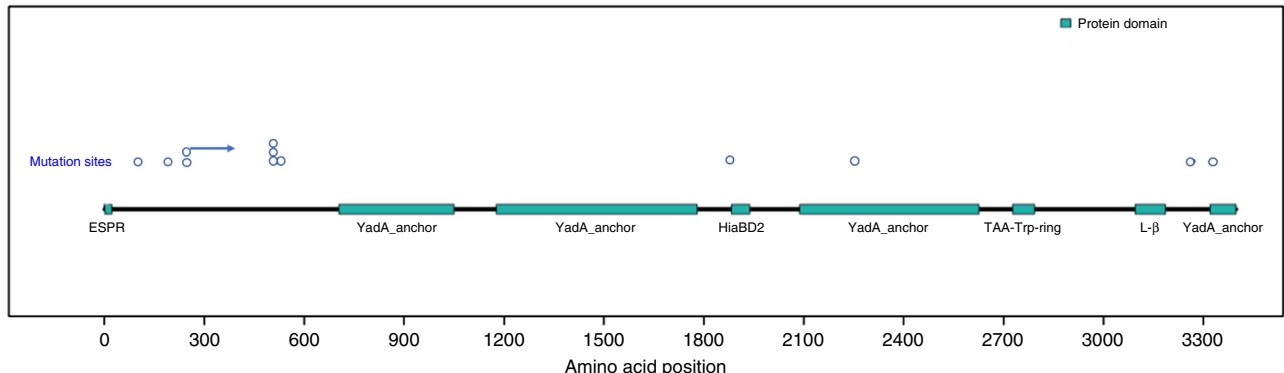

**b** Mutation information

| Volunteer | Timepoint (week) | Mutation type | Mutation annotation | Position | Protein size reduction (%) | Methionine start sites introduced |
|---|---|---|---|---|---|---|
| 172 | 26 | SNP | E102K (GAA→AAA) | Coding (304/10197 nt) | 0 | 0 |
| 172 | 16 | Deletion | Δ997 bp | Coding (0574-1570/10197 nt) | 94 | 1 |
| 158 | 8 | Deletion | Δ461 bp | Coding (0747-1207/10197 nt) | 93 | 0 |
| 158 | 26 | Deletion | Δ461 bp | Coding (0747-1207/10197 nt) | 93 | 0 |
| 227 | 4 | Deletion | Δ1 bp | Coding (1539/10197 nt) | 84 | 1 |
| 227 | 8 | Deletion | Δ1 bp | Coding (1539/10197 nt) | 84 | 1 |
| 227 | 16 | Deletion | Δ1 bp | Coding (1539/10197 nt) | 84 | 1 |
| 315 | 16 | Insertion | GGGGG→GGGGGG | Coding (1577/10197 nt) | 84 | 1 |
| 113 | 16 | Deletion | Δ1bp (AAAAA→AAAA) | Coding (5638/10197 nt) | 45 | 1 |
| 221 | 26 | Deletion | Δ1 bp | Coding (6762/10197 nt) | 34 | 0 |
| 201 | 26 | Insertion | +GG | Coding (9803/10197 nt) | 2 | 5 |
| 104 | 16 | Deletion | Δ1 bp | Coding (9990/10197 nt) | 2 | 5 |

**Fig. 2** Domains, mutations and mutation areas of effect introduced in Large-Hypothetical Protein (ARB05049.1) (**a**) with the distribution found between volunteers (**b**). Mutations are coloured blue and white and are listed in **b** in the order they occur along the protein depicted in **a**

---

**Table 3 Relative recombination effect (r/m) of isolates in colonised volunteers and the parameters used to infer them**

| Colonisation | Organism | ST | Isolates (n) | R/theta | Delta[a] | Nu | r/m |
|---|---|---|---|---|---|---|---|
| N. lactamica | N. lactamica (I) (Y92-1009) | 613 | 97 | 0.48 (5.7 E$^{-4}$) | 35 (2.8 E$^{-2}$) | 0.22 (1.6E$^{-5}$) | 3.61 |
| Co-colonised | N. lactamica (I) (Y92-1009) | 613 | 19 | 0.40 (9.1 E$^{-3}$) | 138 (4.5 E$^{-6}$) | 0.11 (9.1 E$^{-4}$) | 3.84 |
| | N. lactamica (WT) | 11524 | 3 | 0.68 (3.0 E$^{-2}$) | 585 (2.1 E$^{-7}$) | 0.05 (6.4 E$^{-6}$) | 18.06 |
| | N. meningitidis (WT) | 41 | 3 | 0.38 (1.8 E$^{-3}$) | 403 (7.4 E$^{-7}$) | 0.07 (1.2 E$^{-6}$) | 10.69 |
| | N. lactamica (WT) | 4192 | 6 | 0.39 (9.1 E$^{-3}$) | 123 (3.5 E$^{-6}$) | 0.19 (1.4 E$^{-4}$) | 9.21 |
| | N. meningitidis (WT) | 10457 | 3 | 0.28 (1.2 E$^{-2}$) | 420 (9.4 E$^{-7}$) | 0.07 (4.3 E$^{-5}$) | 8.32 |

The table displays recombination parameters among wild-type (WT) N. lactamica and N. meningitidis in addition to inoculated (I) N. lactamica Y92-1009 in single and co-colonised volunteers. The parameters include the relative rate of recombination to mutation (R/theta), the mean length of detected recombinant regions (Delta) and mean divergence level between recipient and donor (Nu). Standard deviations are indicated for these values in the accompanying brackets (X.X E$^{-X}$). The parameters were multiplied together to generate the relative effect of recombination compared to the relative effect of mutation (r/m). The sequence type (ST) of each collection of organisms is also given
[a]Delta is calculated by ClonalFrameML as 1/Delta. This has been resolved in the table to show clearly how r/m was calculated. Standard deviations for this value are for 1/Delta

---

over the longest period (26 weeks) is the most accurate, the value of $2.21 \times 10^{-06}$ substitutions per site, per year is in agreement with estimates detected in N. gonorrhoeae[20–22] and N. meningitidis[23]. Our values for effective population size in the four sample sets (long-term carriage of N. lactamica with and without co-carriage of N. meningitidis; short-term carriage in vivo; in vitro passage) were much lower ($N_e = 1$–25) than those recently described among Escherichia coli examined over a year[24], but comparable to those observed in Vibrio cholerae[25]. These latter two studies, however, examined different organisms in different niches, with the Vibrio observations collected during acute infection. A low effective population size may indicate the presence of severe bottlenecks or growth limitation, and our

observations may be an outcome of the dynamic antagonistic and commensal processes that occur among members of the microbiota of the upper respiratory tract[26]. It is worth noting, too, that, at least in murine models of pneumococcal colonisation, $N_e$ was also found to be low, between 11 and 203, and thus comparable to our findings [27,28]. Although these numbers are derived from single colony isolates, and thus may not reflect the true genomic heterogeneity of the population [29] the low number of mutations observed longitudinally within individuals suggests the estimation of both mutation rate and Ne are accurate. Moreover, a similar substitution rate of $1.45 \times 10^{-5}$ SNPs per site per year was observed when multiple colonies were sampled among six volunteers in the short-term study (1 month), this is in contrast to

**Table 4 Genes undergoing recombination**

| Organism | Volunteer | Recombinant NEIS loci | Gene alias | Annotation | Direction of recombination |
|---|---|---|---|---|---|
| *N. lactamica* (I) | 166 | NEIS1708 | NMB0444 | hp | Unknown |
| *N. lactamica* (I) | 133 | NEIS1794 | N/A | Putative mafS3 cassette | Unknown |
| *N. lactamica* (I) | 38 | NEIS1828 | NMB0342 | Intracellular septation protein A | Unknown |
| *N. lactamica* (WT) | 36 | NEIS1795 | NMC1795 | MafI immunity gene: type o2MGI-2 | Unknown |
| *N. lactamica* (WT) | 291 | NEIS0069 | NMB0085 | Sodium glutamate transport | Unknown |
| *N. lactamica* (WT) | 291 | NEIS0073 | NMB0088 | Outer membrane transport | Unknown |
| *N. lactamica* (WT) | 291 | NEIS1837 | NMB0334 (pgi2) | Glycolysis pathway enzyme | Unknown |
| *N. lactamica* (WT) | 291 | NEIS2437 | NMB0336 | Putative lipoprotein | Unknown |
| *N. lactamica* (WT) | 291 | NEIS0071 | NMB0071 | Putative lipoprotein | Interspecific |
| *N. lactamica* (WT) | 291 | NEIS0072 | NMB0087 | hp | Interspecific |
| *N. lactamica* (WT) | 291 | NEIS0899 | NMB0922 (lst) | LOS alpha-2,3-sialyltransferase | Interspecific |
| *N. lactamica* (WT) | 291 | NEIS2514 | hp | — | Interspecific |
| *N. lactamica* (WT) | 291 | NEIS1834 | NMB0336 | Enoyl-(acyl carrier protein) reductase | Interspecific |
| *N. lactamica* (WT) | 291 | NEIS1835 | NMB0335 | N-succinyltransferase | Interspecific |
| *N. meningitidis* (WT) | 36 | NEIS2149 | NMB0011 | Peptidoglycan (cell wall) biosynthesis | Unknown |
| *N. meningitidis* (WT) | 36 | NEIS2859 | hp | — | Unknown |
| *N. meningitidis* (WT) | 291 | NEIS0210 | NMB0018 (pilE) | Pilus component synthesis | Unknown |
| *N. meningitidis* (WT) | 291 | NEIS1402 | hp | — | Unknown |

The table lists genes identified as recombinant in the dataset following analysis by ClonalFrameML. Isolates of *N. lactamica* are differentiated with the labels wild-type (WT) and artificially-inoculated (I). Gene names are given as found on PubMLST *Neisseria* (NEIS loci), in addition to more commonly recognised gene aliases. The label interspecific, refers to loci that shared the same recombinant allele with their co-colonising species during the course of the study

the finding of a substitution rate of $2.21 \times 10^{-6}$ SNPs per site per year in the long-term study (6 months) across many more volunteers, albeit over a much shorter period of colonisation. Additionally, the bulk (70%) of the mutations observed in the short term in vivo (Supplementary Data 1) cohort occurred once per colony sampled. Mutation prevalence over all colonies at a given time point and mutation recurrence across time points were observed to be rare events.

Purifying selection acts to remove non-beneficial mutations, and our observation that most SNPs were transient, non-recurrent events, combined with a negative Tajima's D value, suggest that diversity was lower than expected, consistent with the population being subject to bottleneck events. Despite this, some non-synonymous SNPs persisted longitudinally in some volunteers, becoming fixed in the sample and staying for at least one further time point. These included non-novel mutations in *rssA* and novel mutations in the *nuoN*, *rssA*, 2 hypothetical proteins and *cas1* loci. The *nuo* operon, which encodes subunits for NADH dehydrogenase complex I, is among those genes upregulated in iron replete conditions in *N. gonnorhoeae*[29] and *N. meningitidis* incubated in human blood[30]. Cas1 acts as a dsDNA endonuclease in a minimal, self-promoted, CRISPR-Cas type II-C system in *N. lactamica*, that has been postulated to provide adaptive protection to its host from unbeneficial, exogenous, horizontal gene transfer[31]. The persistent mutations in phospholipase RssA represent the first evidence of this gene adapting to the oropharyngeal environment.

Most of the insertion and deletion events not present in contingency loci affected one gene: a large, (~10,000 bp) multi domain hypothetical protein (L-hp). An in-silico simulation of how these mutations might affect protein coding revealed frame-shifted introductions of interproteomic methionine residues and overall reductions in predicted protein size. The L-hp locus was the only region in which mutations were seen to persist in some volunteers. The low GC content of this locus relative to *N. lactamica* Y92-1009 suggests that it may have been horizontally acquired by *N. lactamica* from another nasopharyngeal coloniser, perhaps *H. influenzae*, based on the results from CDART and the

possession of a *H. influenzae* autotransporter domain. This is not uncommon and there are other examples of genes postulated to have been acquired by *N. lactamica* 020-06 from *H. influenzae*, including phosphroylcholine biogenesis genes *licABCD*, ATPase gene *slpA*, protease gene *slpB* and putative surface fibril protein NLA12600[32]. The presence of trimeric autoadhesins such as the YadA -like and Hia conserved domains, suggest this protein may have a role in initial attachment of the bacterium to a human host[33]. Post-colonisation, it may be that the deletion of these proteins is beneficial, perhaps metabolically or as it is a site for host-recognition.

In conclusion, *N. lactamica* Y92–1009 possesses a stable genome that undergoes stochastic mutation, at a low mutation rate with minimal evidence of recombination or interspecific interaction with co-colonising meningococci, as was demonstrated by wild-type *N. lactamica*. However, the observed genetic changes indicate an important role of phase variation among genes associated with host-commensal interaction and iron acquisition in sustained colonisation of human hosts. A lack of microevolutionary evidence in *N. lactamica* isolates co-colonised with *N. meningitidis* suggests that the meningococcal carriage suppression induced by *N. lactamica* may be due to outcompeting the pathobiont for essential resources in the nutrient-limited, nasopharyngeal niche.

## Methods

**Controlled human infection with *N. lactamica*.** *N. lactamica* Y92–1009 was originally isolated during a 1992 carriage study of school pupils in Londonderry, Northern Ireland. Its use in an experimental human challenge model (Clinical Trial Registration: NCT02249598) has been reported previously [16]. The study was approved by the National Health Service Research Ethics Committee (11/YH/0224), was overseen by an independent safety committee, and was conducted in accordance with the Declaration of Helsinki. Written informed consent was gained from participants prior to inclusion in the study. Briefly, 149 participants received nasal challenge with $10^4$ CFU of *N. lactamica*. In this group, natural *N. meningitidis* carriage at baseline was 36/149 (24.2% [95% CI, 17.5%–31.8%]). Natural carriage of *N. lactamica* prior to inoculation was observed in 3/149 individuals (1.9% [95% CI, .4%–3.5%]). Following challenge, oropharyngeal swabs were taken at 2, 4, 8, 16, and 26 weeks and were plated directly onto gonococcus (GC)-selective media (E&O Laboratories, Scotland) and incubated at 37 °C in 5% $CO_2$. After 48 hours,

possible *Neisseria* spp. colonies were subjected to API-NH strip testing (bioMérieux, France) and PCR[16]. The isolates were not extensively passaged. Growth was limited to two passages (one for isolation from the host and one for preparation of sequencing). Two weeks after inoculation, oropharyngeal swabs from 48 individuals challenged with *N. lactamica* yielded cultivable *N. lactamica* (33.6% [95% CI, 25.9%–41.9%]), but colonization with *N. lactamica* was detected as late as week 26, by which time colonisation with *N. lactamica* at some point in the study had been confirmed in 61 of 149 (41.0% [95% CI, 33.0%–49.3%]) of the challenge group. All isolates of *N. lactamica* and *N. meningitidis* were cryostored. The study flow is summarised in Supplementary Figure 3.

**Choice of *Neisseria* isolates for genomic analysis**. The inclusion strategy prioritised the following: (a) isolates from individual volunteers that were recovered from multiple sampling points; (b) isolates from volunteers co-colonised with *N. meningitidis*; and, (c) isolates recovered either early (week 2) in the study or later (week 26), with the rationale that the isolates from earlier time points had undergone adaptation and selection more recently and the isolates at the late time point would have a greater chance of demonstrating signs of convergent evolution, having colonised their respective hosts for the longest period of time. In total 97 *N. lactamica* Y92–1009, 14 *N. meningitidis* and 9 non-Y92–1009 *N. lactamica* isolates were chosen for sequencing and analysis. These isolates are summarised in Supplementary Data 6

**Comparison of in vitro passage vs. in vivo colonisation**. To compare stochastic mutation during in vitro passage with variation observed intra-host, 10 additional volunteers were intra-nasally inoculated with $10^4$ CFU ml$^{-1}$ *N. lactamica* Y92–1009 stock. Oropharyngeal swabs were used to take samples at days 1, 2, 3, 14 and 28. Colonisation was established in five of the ten volunteers. In vitro serial passages were done in duplicate and in parallel with the in vivo colonisation study. Isolates, consisting of 1–5 colonies, were subcultured on Columbia blood agar every weekday for 28 days. A maximum of five colonies of *N. lactamica* from each sampling point were selected and sequenced.

**Culture**. Bacterial samples were recovered from cryostorage (−80 °C) and streaked onto Columbia + horse blood plates and cultured overnight at 37 °C. *N. lactamica* colonies were confirmed using X-gal (Sigma Aldrich, USA) in PBS. Two blue colonies were then subcultured into bijoux tubes, each containing 2 ml Tryptic Soy broth with 0.2% yeast extract (Sigma Aldrich, USA). The cultures were grown overnight with agitation at 37 °C.

**Genomic DNA extraction**. Genomic DNA was extracted using the Wizard Genomic Purification Kit (Promega, USA). Sample purity and concentration (OD 260/280) were assessed using a Nanodrop 1000 spectrophotometer (ThermoFisher, UK) and Qubit 2.0 fluorometer with the BR dsDNA kit (Invitrogen, UK). For long read sequencing of the reference genome, a sample of cell bank purified stock was cultured and extracted using the Gentra Puregene yeast/bacteria kit according to manufacturer's instructions (Qiagen, Germany) to produce 30 μg of high molecular weight (>40 kDA) DNA.

**Sequencing**. All isolates of *N. lactamica* were sequenced using an Illumina HiSeq 2000 instrument as described previously[34] to generate 2 × 151 bp PE libraries. For the reference genome, high molecular weight DNA was pooled and sequenced using the Pacific Biosciences RSII instrument as outlined in[17] (Earlham institute, Norwich). Data for the reference strain is available on NCBI with accession CP019894.

**Genome assembly, correction and annotation**. Reads were assembled using velvet optimiser and added to the PubMLST Neisseria database[35] [https://pubmlst.org/neisseria/]. The resultant contigs were annotated using automated processes to identify specific loci using the PubMLST.org/neisseria 'NEIS loci' catalogue[36] The reference genome was assembled at the Earlham institute (as described in[17]). Error correction of the reference was done with Illumina reads using Pilon v1.17[37]. Annotation was done using Prokka v1.11[38] aided by a curated protein list from *N. lactamica* 020-06 (accession: FN995097).

**Mutational and recombination analyses**. Mutational analysis was performed by mapping reads against the reference genome assembly using the breseq v0.23b pipeline[39]. In addition, in order to determine the novelty of persistent, volunteer-specific, and in one case, non-specific non synonymous mutations identified in this study, the specific location and amino acid changes relative to that position of mutants were searched via sequence database BLAST of the entire *Neisseria* loci catalogue hosted on pubMLST *Neisseria* database and via non-specific BLASTp against the nr database.

Recombination analysis was done using ClonalFrameML v1.25 (CFML)[40] using core XMFA alignments and a core genome, maximum-likelihood (ML) tree as input. The genome comparator tool [https://pubmlst.org/neisseria/]was used to determine core NEIS loci. Before generating an alignment, the NEIS loci were exported in FASTA format and screened for paralogues using CD-HIT v4.6.6[41]

which were excluded from the pool of core NEIS loci. The genome comparator tool was then used to output core XMFA and FASTA alignments, the former to be used as input to CFML, the latter to be used to construct a ML tree. The ML tree was created by converting the FASTA alignment to PHYLIP format using EMBOSS seqret v6.6[42] and running PHYML v20131022[43] on the PHYLIP alignment using the HKY85 model. This process also outputted a transition/transversion ratio to be used as the -kappa input parameter in CFML. ML trees were converted to binary format using the R package ape v3.0[44]. The use of XMFA files with CFML introduced 1000 bp spacer segments between each gene in the alignment. The subsequent data on the location of recombinant alleles was adjusted to correctly confirm their identity. Any recombinant alleles were examined manually using mView[45] to confirm significant divergence suggesting recombination instead of false positives due to interspersed, minor mutation events.

**Phase variation ON VS OFF statistics**. Six genes, *lgtG*, *lgtC*, *pglA*, *pglH*, *hsdS* and *hpuA* were tested for significant differences in phase ON vs phase OFF status in weeks 2, 4, 8, 16 and 26. These six were tested as we limited the analysis to those that underwent at least one phase variation and where putative ON/OFF expression status data was available in the literature from other *Neisseria* spp.[46,47]. In addition, *fetA* was excluded as this is not an ON/OFF phase variable event but a promotor-dependent modulator for transcriptional efficacy, the role of which is unconfirmed in this *Neisseria* spp. Fishers exact tests using 2 × 2 contingency tables were tested in GraphPad Prism contingency table calculator[48]. The input into the table was the number of isolates for an early time point which had putative phase ON copy numbers vs putative phase OFF copy numbers. The second row of the contingency table contains the isolates with ON/OFF copy number for the later time point to be tested. When testing isolates in week 2, null values were used in place of week zero samples in the contingency table.

**Protein sequence analysis**. Where applicable, hypothetical proteins were investigated using the conserved domain architecture tool[49], which was used to meta-analyze the predicted protein domains. These domains, alongside any mutations that were seen to occur during longitudinal carriage, were used as input for plot protein[50] to generate a protein diagram. In silico translations to determine effects of mutation on protein size were examined using ExPASy translate[51].

**Effective population size, selection and mutation rate calculations**. Core loci alignments, generated as described above, were used as input for the R package PopGenome[52] to test for selection using Tajima's D[53] and to calculate diversity via Tajima's theta ($\theta$). The mean values for Tajima's D and $\theta$ were taken and the standard deviation of the population was calculated. Mutation rate ($m$) was calculated as the number of SNPs per site per year. Effective population size ($N_e$) was estimated by rearranging the formula $\theta = 2gN_emL$ to $N_e = \theta / (2mgL)$[54], where $\theta$ is the mean genetic diversity using Tajima's theta, $m$ is the mutation rate per site per year, $g$ is the generation rate (estimated to be roughly one cell division per day: 1/365) and $L$ is the genome length (2,146,723 bp).

## Data availability
The complete reference genome is available[17] as published work and on GenBank (accession: CP019894.1). One of the major proteins of interest dubbed "l-hp" can be found at accession number ARB05049.1. Sample assemblies and reads from the long-term study have been deposited in the European Nucleotide Archive (ENA) and can be accessed using the study access code PRJEB9839. A full list including volunteer codes, isolates, and ENA accession codes is shown in Supplementary Data 7. Sample assemblies and reads can also be accessed by searching for the "ID number" (displayed in Supplementary Data 7) in the PubMLST Neisseria isolate search page [https://pubmlst.org/bigsdb?db = pubmlst_neisseria_isolates&page = query]. The database can be queried in the following manner. The Isolate Provenance Field must be changed to "ID". The value in the middle must be left as an " = " sign. The ID number can be now entered and submitted. Other data supporting the findings of the study are available in this article and its Supplementary Information files, or from the corresponding authors upon request.

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

## Acknowledgements

This project was supported by funding from the UK Medical Research Council: MR/N013204/1 and MR/026993/1, The Sir Christopher Benson Studentship (University of Southampton), the Wessex Institute of Virology and Infectious Disease and the Southampton NIHR Biomedical Research Centre. This publication made use of the PubMLST website [https://pubmlst.org/] developed by Keith Jolley and Martin Maiden[35] and sited at the University of Oxford. The development of that website was funded by the Wellcome Trust. We also made use of the iridis4 compute cluster [https://www.southampton.ac.uk/isolutions/staff/iridis.page] hosted at the University of Southampton and wish to thank the HPC support staff in particular, Dr Elena Vataga.

## Author contributions

A.P. performed the analyses described within and wrote the first draft of the manuscript. D.W.C. suggested further analyses and alongside R.C.R. helped construct the manuscript and revisions. J.R.L., A.G., X.D. and M.C.J.M. helped with study design, analysis and made suggestions/corrections to the manuscript. P.D.M., A.P.D., A.M.D. and R.C.R.

conceived and executed the two clinical studies from which the study isolates were sourced.

## Additional information

**Competing interests:** The authors declare no competing interests.

