## [Peer Review File · Nature Communications]

Reviewers' comments:

Reviewer #1 (Remarks to the Author):

Panday et al studied used banked isolates to examine sequence changes that occurred during *N. lactamica* experimental human colonization. Volunteers were inoculated with a strain of *N. lactamica* and isolates were taken at various time-points over 26 weeks. A short-term experiment was also done with five volunteers over a month in which in vivo evolution was compared to laboratory grown isolates. The genomic sequences of these isolates revealed DNA sequence changes in the isolates over this time period. Notably, the repetitive tracts of contingency loci harbored the most insertions or deletions. Also, one hypothetical protein was deleted in multiple volunteers, suggesting a selective pressure for this deletion. The retrospective nature of this work present issues with the experimental design. Perhaps because of this issue, there are no strong conclusions that can be drawn from this work, except that there are sequence changes in these short time scales. Finally, there are major issues with presentation that make it very difficult to determine what was done and what the data might mean. The study of within-host evolution is under explored and while it is important to undertake these studies to understand the process of colonization this work does not significantly add to our understanding of this organism or its relationship to its host.

Major concerns

It appears that single re-isolates were obtained from volunteers at various time points throughout this study (the inoculating strain was a human isolate originally). However, there is no mention of the fact that each isolate may not be representative of the heterogeneous population of *N. lactamica* present in each volunteer. *Neisseria* spp. have many phase variable loci, which is mentioned in the paper, but the authors should discuss that this study is limited because the *Neisseria* population at each time-point is most likely heterogeneous and each sampling maybe be a different isolate of the population and/or the population is changing (Summarized for *N. gonorrhoeae* and *N. meningitidis* in Hobbs, MM et al *Frontiers in Microbiology*, 2011). Moreover, it is not clear how many times these re-isolates were passaged before the sequencing was performed (Were they really grown for 48 hours?) and no consideration given to the possibility that some of the changes occurred after re-isolation. A frank discussion of the potential limitations of the data is needed

The authors conclude that they can detect recombination between *N. lactamica* and *N. meningitidis*. However, this was not seen with the lab inoculated strain and only a naturally occurring colonization. It is therefore impossible to conclude these recombination events happened during the time frame of this study but that is implied.

The increase in phase variation within the host re-isolated organisms is potentially interesting and some of the altered gene functions are discussed. It would be interesting to better define or speculate the outcome of the phase variation. Was there a noticeable pattern in which certain genes were turned OFF or ON, such as in the short term experiment or the genes in Figure 1B?

Minor comments

Line 27: I'm not sure commensal bacteria, "must resist host immunity" if they are not perceived as foreign.

Lines 44-45: Instead of "wild-type", consider "naturally acquired"?

Line 47: "This ___" refers to what?

Line 58: Consider referencing Richardson, *Proc Natl Acad Sci U S A*. 2002 Apr 30;99(9):6103-7.

Line 80: It would be accurate to say that you are exploring the potential protective affect of *N. lactamica*.

Line 92: The length of the short term experiment should be mentioned in this section.

Line 94: A short description of what these isolates are and how they were obtained and whether they were passaged should be addressed here. It is also important to point out that this inoculated strain is adapted for laboratory growth.

Line 106: Note the *phrB* gene of *Neisseria gonorrhoeae* was found not to be a photolyase (Cahoon, *Mol Microbiol.* 2011 Feb; 79(3): 729-42).

Line 111: The term phasevarion has been used to describe phase variation of a methylase that controls transcription of many genes. Is this what is meant here? (e.g., Atack, *Trends Microbiol.* 2018 Feb 13).

Lines 120-121: Were generations or growth rate used in the calculation of substitution rate?

Line 122. A description of what Tajima's coefficients are and what the values reported mean would be helpful.

Table 1: Not clear what "Shared Genotype" means and possibly this should be defined in a footnote. Unclear what intragenic denotes and whether as the text indicated these are all within promoter sequences.

Table 2: it is unclear whether these values are really difference and whether there are statistical test that should be applied here.

Lines 202-205. Are these all predicted to be loss of function mutations? Is the deletion or modification of this hypothetical protein novel? Has it been seen in any other *N. lactamica* isolates?

Lines 222-225. Why only having one isolate was an issue is not obvious.

Figure 1: The color scale is for both A and B.

Figure 2. What is the difference between white and blue rows?

Figure 3. What is the "Isolates" column? 1154 isolates?

Lines 283-285 What do the population sizes mean in respect to other bacteria in the same niche?

Lines 322-25. This conclusion does not seem to follow from the data.

Supp Table 2 What are the numbers in the blue and purple boxes? The SNP difference for each allele in the first column is confusing. Also, *N. lac* should not be capitalized.

Reviewer #2 (Remarks to the Author):

In the manuscript entitled "Microevolution of *Neisseria lactamica* during nasopharyngeal colonisation induced by controlled human infection", Pandey and colleagues capture and classify genomic mutation in *N. lactamica* during human nasopharyngeal carriage. Understanding of

microevolution of human pathogens in their natural habitat is critical for the long-term success of vaccines and therapeutics. This study stands out for the use of human volunteers; there are very few such studies.

The author's calculate the rates of mutation and recombination, and their findings suggest a limited number of these events during the single and multi-strain infections. Further, their findings reveal that the main mechanism of mutation during colonization is phase variation. In addition, they identify genes that were mutated in longitudinal samples and/or multiple individuals.

Comments:

(1) For the multi-species volunteers (7) the r/m is calculated at 3.84 (line 225), and the parameters used to calculate this ratio are delineated in Table 3. However, I understand the sentences after line 225 to state that there were no recombination events in this set. Please clarify.

(2) For volunteer 291, the r/m ratio is 18. Since these strains were both naturally co-colonizing this volunteer, it is possible that they exchanged DNA before the experiment start point. Please add a description of what was used as a reference for r/m ratios. Please clarify and note caveats associated with the r/m calculation in this case.

(3) For the persistent mutations, what is their distribution across the species?
That is, are the polymorphisms acquired during infection present in other sequenced isolates?

(4) The number in lines 140-147 do not agree with those in Fig. S3. Please correct or clarify.

(i) It appears as 97 versus 95 for strain Y92-1009, and 53 versus 32 for monoclonal colonization.

(5) Please provide more detail on the isolation of strains from the swabs. For instance:

(i) on average, how many colonies were present on the plated oropharyngeal swabs?

(ii) how was *N. lactamica* and *N. meningitidis* separated from other *Neisseria* species? Was it visual or was any additional molecular testing performed on the colonies?

(6) Line 266-273: Does this imply independent transfer events into the *N. lactamica* strain? One set captures in the initial collection and one set later? Please expand.

(7) The sequence similarity analysis suggests that L-hp is shared across strains in this species. The conclusion drawn that it is a HGT from *H. influenzae* requires more data. At this point the analysis simply suggests these are similar sequences, and these could be transferred across these species. At a minimum, does the GC content provide additional support for *H. influenzae* as a donor?

(8) 291-302: This paragraphs implicate neutral and adaptive mutations for *rssA* and other persistent mutations. Please clarify language.

Very minor:

Lines 221-225. Sentence is confusing to read. Please write more clearly specifying co-carried.

Add a period to the end of the sentences in Discussion.

Response to Reviewers

Re: Pandey et al ; NCOMMS-18-14032T: Microevolution of *Neisseria lactamica* during nasopharyngeal colonisation using an experimental human challenge model

Reviewer #1 Major Comments

1. The retrospective nature of this work present issues with the experimental design. Perhaps because of this issue, there are no strong conclusions that can be drawn from this work, except that there are sequence changes in these short time scales.

Authors response: We respond that this unique study, with follow up of 6 months of human colonization, provides very strong evidence to support the conclusions; detailed analysis of sequence changes of a human-adapted colonist during a prolonged period of colonization in multiple volunteers. Our comparison of *in vivo* and *in vitro* sequence modifications reassures us that the changes we have observed during colonization of humans are not stochastic. We have utilized state of the art, quantitative measures such as effective population size and mutation rate and have been able to examine population genetics, microevolutionary pressure and the impact of recombination in intra-host environments that are also colonized by another *Neisseria sp.* The conclusions we draw are supported by the data and make sense.

2. There are major issues with presentation that make it very difficult to determine what was done and what the data might mean. The study of within-host evolution is under explored and while it is important to undertake these studies to understand the process of colonization this work does not significantly add to our understanding of this organism or its relationship to its host.

Authors response: We appreciate the reviewers concerns and would like to address them in two parts.

Firstly, in terms of presentation, we believe that we have addressed all the subsequent issues raised by the reviewer and feel that what we have done is clear and interpretable. Secondly, we respectfully disagree with the reviewer's statement that this work "...does not significantly add to our understanding...". Studies on intra-host microevolution are rare, and this study, in that respect alone, represents a progression in our understanding of human colonisation..

The experiment we are seeking here to report is one of the few experimental human challenge models and certainly the information we provide has not been provided before, **for any colonist of the upper respiratory tract.**

3. It appears that single re-isolates were obtained from volunteers at various time points throughout this study (the inoculating strain was a human isolate originally). However, there is no mention of the fact that each isolate may not be representative of the heterogeneous population of *N. lactamica* present in each volunteer. *Neisseria spp.* have many phase variable loci, which is mentioned in the paper, but the authors should discuss that this study is limited because the *Neisseria* population at each time-point is most likely heterogeneous and each sampling maybe be a different isolate of the population and/or the population is changing (Summarized for *N. gonorrhoeae* and *N. meningitidis* in Hobbs, MM et al *Frontiers in Microbiology*, 2011).

Authors response: The reviewer's comment is useful and in reflection of this we have amended the discussion section of the manuscript to further highlight the potential confounders of isolate selection {lines 312-322}. We would like to highlight however that our data suggests a small population size post-inoculation and a low mutation rate with near-identical isolates within individuals at different time points. We therefore posit that the impact of heterogeneity, whilst important to note, does not limit the interpretation of our findings.

4. It is not clear how many times these re-isolates were passaged before the sequencing was performed (Were they really grown for 48 hours?) and no consideration given to the possibility that some of the changes occurred after re-isolation. A frank discussion of the potential limitations of the data is needed

Authors response: The reviewer raises an important issue, but one we have considered and are happy to reflect in the discussion {lines: 372-373}. The isolates were not extensively passaged. Growth was limited to two passages (one for isolation from the host and one for preparation of sequencing). We accept that these two periods of growth on selective media may have introduced changes, however we argue (i) that, pragmatically, this is unavoidable in such studies and (ii) the number, type and location of mutations seen in the *in vitro* passage isolates were sufficiently different to allow us to interpret our *in vivo* findings accurately.

5. The authors conclude that they can detect recombination between *N. lactamica* and *N. meningitidis*. However, this was not seen with the lab inoculated strain and only a naturally occurring colonization. It is therefore impossible to conclude these recombination events happened during the time frame of this study but that is implied.

Authors response: We completely agree with the reviewer's comments. We certainly did not intend to imply this recombination occurred during the study. We have amended the manuscript to include this caveat {lines 279-283}.

6. The increase in phase variation within the host re-isolated organisms is potentially interesting and some of the altered gene functions are discussed. It would be interesting to better define or speculate the outcome of the phase variation. Was there a noticeable pattern in which certain genes were turned OFF or ON, such as in the short-term experiment or the genes in Figure 1B?

Authors response: We thank the referee for this suggestion and have undertaken this evaluation. The following has been added to the manuscript in the methods section {lines 434-446}. And results section {lines 113-117}

‘Six genes, *lgtG*, *lgtC*, *pglA*, *pglH*, *hsdS* and *hpuA* were tested for significant differences in phase ON vs phase OFF status in weeks 2, 4, 8, 16 and 26. These six were tested as we limited the analysis to those that underwent at least one phase variation and where putative ON/OFF expression status data was available in the literature from other *Neisseria* spp. (Zelevska *et al*, 2016, Snyder *et al.*, 2001). In addition, *fetA* was excluded as this is not an ON/OFF phase variable event but a promotor-dependent modulator for transcriptional efficacy, the role of which is unconfirmed in this *Neisseria* spp. Fishers exact tests using 2 × 2 contingency tables were tested in GraphPad Prism.’

As shown in supplemental data 6, 2 × 2 contingency tables were constructed and tested with Fisher's exact test. The input into the table was the number of isolates displaying phase on/off at each time point compared to those at an earlier time point (or in the case of week 2 isolates, to time point 0). This was done to determine whether there was a significant accumulation of a particular phased genotype

(supplementary data 6). The outcome tested therefore was a change of genotype respective of the last timepoint. Of the six genes tested, only *lgtG* and *hpuA* were significantly different ($p < 0.05$) at weeks 8 and 4 respectively. This represented a shift from phase ON to phase OFF.

So, we state:

`No consistent pattern of phase switching ON vs phase switching OFF was observed among the isolates tested (supplemental data 6)`, however, two genes, the haemoglobin receptor *hpuA* and glycosyltransferase *lgtG* exhibited a significant shift from PHASE ON -> OFF in weeks 8 and 4 respectively ($p=0.05$ Fisher's exact test).`

Reviewer #1 Minor Comments

1. Line 27: I'm not sure commensal bacteria, "must resist host immunity" if they are not perceived as foreign.

Author response: We have changed the introductory sentences.

2. Lines 44-45: Instead of "wild-type", consider "naturally acquired"?

Author response: Changed as suggested.

3. Line 47: "This ___" refers to what? ___

Author response: We are unsure what the reviewer is referring to. The section below relates to lines 44:47. "To summate, the study shows that mutation is prominent among known and putative phase variable regions; that interpersonal variation is present with strong colonisation bottlenecks and recombination drives genetic diversity, but only among volunteers with an established, natural, co-carriage of *N. meningitidis* and *N. lactamica*."

4. Line 58: Consider referencing Richardson, Proc Natl Acad Sci U S A. 2002 Apr 30;99(9):6103-7.

Author response: We thank the reviewer for highlighting this and we have referenced accordingly.

5. Line 80: It would be accurate to say that you are exploring the potential protective affect of *N. lactamica*.

Author response: That is indeed correct and we have adjusted the wording in line with the reviewer's comment.

6. Line 92: The length of the short-term experiment should be mentioned in this section.

Author response: We thank the referee for noticing this detail and have altered accordingly.

7. Line 94: A short description of what these isolates are and how they were obtained and whether they

were passaged should be addressed here. It is also important to point out that this inoculated strain is adapted for laboratory growth.

Author response: The manuscript has been amended to include the following text “We recovered isolates from throat swabs taken from volunteers who had been inoculated with laboratory strain *N. lactamica* 10⁴ CFU ml⁻¹ cell bank stock Y92-1009 [17]”. {Lines 86, 87}

8. Line 106: Note the *phrB* gene of *Neisseria gonorrhoeae* was found not to be a photolyase (Cahoon, Mol Microbiol. 2011 Feb;79(3):729-42).

Author response: Thank you to the reviewer for pointing this out. We have altered the text to reflect the re-annotation of this ORF to *phrB* and have added the following statement and referenced the Cahoon *et al* study. “confers resistance to nalidixin and rifampicin in *N. gonorrhoeae* [18]”. {lines 98, 99}

9. Line 111: The term phasevarion has been used to describe phase variation of a methylase that controls transcription of many genes. Is this what is meant here? (e.g., Atack, Trends Microbiol. 2018 Feb 13).

Author response: We apologise for the confusion. We were not referring to the phasevarion (*modA*) and have altered its use in the text.

10. Lines 120-121: Were generations or growth rate used in the calculation of substitution rate?

We did not use generations or growth rate in the calculation of substitution rate. These estimates are based on the number of substitutions observed between two serially isolated genomes separated by a known amount of time.

11. Line 122. A description of what Tajima’s coefficients are and what the values reported mean would be helpful.

Authors response: We have added a short descriptive sentence for Tajima’s D and additional text to explain the negative values we identified. We hope that this provides the necessary clarity. {Lines 120-122}

12. Table 1: Not clear what “Shared Genotype” means and possibly this should be defined in a footnote. Unclear what intragenic denotes and whether as the text indicated these are all within promoter sequences.

Authors response: We recognize the lack of explanation here. As suggested we have included footnotes to describe both the (now) “Common Genotypes” and “Location” columns, giving specific reference to intergenic loci.

13. Table 2: it is unclear whether these values are really difference and whether there are statistical test that should be applied here.

We do not claim to have found significant differences between the four sets shown in Table 2. However, we note that all four Tajima’s D values are clearly negative and around -1 and that the estimates effective population size are consistently low Amendments were made to results paragraph {lines 120-127}.

14. Lines 202-205. Are these all predicted to be loss of function mutations? Is the deletion or modification of this hypothetical protein novel? Has it been seen in any other *N. lactamica* isolates?

Authors response: This protein was present in a 44 contig, previously sequenced assembly of the *N. lactamica* Y92-1009 (Score:6597, Identity 100%). Based on a paucity of intact, high-quality matches with blastP (<80 % Identity), and total lack of matches via locus query using the PubMLST Neisseria isolate repository we surmise this is potentially novel to this strain of *N. lactamica* (Y92-1009). Interestingly sections of the protein were present at high identity but only in *Neisseria polysaccharea* (Sequences: 2, Mean Identity: 80.4%, Score: 4346) and *Haemophilus Influenzae* (Sequences: 120, Mean Identity ~50-60%, Score: ~1000-2500). Given the lack of comparators it is difficult to determine the extent to which mutations in similar loci are present. However, the mutations are absent in the 2010 genome assembly and therefore they appear unique to this study. {Please see lines 200-203}

15. Lines 222-225. Why only having one isolate was an issue is not obvious.

Authors response: We have added some clarifying sentences {results section: lines 239-243} to illustrate why isolates from one volunteer were excluded. We don't believe this is a significant issue.

16. Figure 1: The color scale is for both A and B.

Authors response: This has been amended.

17. Figure 2. What is the difference between white and blue rows?

Authors response: This was attempted for visual clarity in differentiating between persistent and singular mutations affecting the protein and its domains. However, we appreciate it may have not had the desired effect. The colours have now been excised.

18. Figure 3. What is the "Isolates" column? 1154 isolates?

Authors response: This is an error in the table, 1154 refers to the volunteer number. We have amended this and added (n) to the table to show this refers to the number of isolates.

19. Lines 283-285 What do the population sizes mean in respect to other bacteria in the same niche?

Authors response: This is an interesting question, but one that has not been studied extensively. Analysis of N_e in murine models of pneumococcal colonisation of the respiratory tract also found a lower than expected population size (compared to census populations). Whether this should be extrapolated to our findings is debatable. However, we have included a line in the discussion to point the reader to this additional reference, and posed the, to our mind open question, about whether this is reflective of the natural state of microbial inhabitants of the URT.

20. Lines 322-25. This conclusion does not seem to follow from the data.

Authors response: We are unsure as to which aspect of our conclusion doesn't follow the data. However, we have revised the final conclusion {lines 343-351} to ensure clarity.

21. Supp Table 2 What are the numbers in the blue and purple boxes? The SNP difference for each allele in the first column is confusing. Also, N. lac should not be capitalized.

Authors response: We have corrected the error and altered the table footnote such that it now reads "Loci undergoing recombination in volunteer 291 with SNP differences between exchanged alleles. Allele cells are colored either blue (early timepoint allele) or purple (later time point allele) to identify them as

distinct for each new allele discovered for the locus. The values in the “SNP difference between alleles” denote the number of variant positions found when contrasting the early timepoint allele to the late timepoint allele. N.M refers to *Neisseria meningitidis*; N.L are *Neisseria lactamica*.” We hope this provides the necessary explanation as requested by the reviewer.

Reviewer #2 Major Comments

1. For the multi-species volunteers (7) the r/m is calculated at 3.84 (line 225), and the parameters used to calculate this ratio are delineated in Table 3. However, I understand the sentences after line 225 to state that there were no recombination events in this set. Please clarify.

Authors response: We apologize for the confusion here. The point we tried to make is that the importations that were identified were small and occurred only in intergenic regions, not that there was no detected recombination. We have removed the final sentence from the paragraph to remove the conflicting narrative. {lines 243-246}

2. For volunteer 291, the r/m ratio is 18. Since these strains were both naturally co-colonizing this volunteer, it is possible that they exchanged DNA before the experiment start point. Please add a description of what was used as a reference for r/m ratios. Please clarify and note caveats associated with the r/m calculation in this case.

Authors response: The reviewer is absolutely correct. We have amended the text to highlight this important caveat of the data. {lines 279-283}. Regarding the issue of reference, the approach we use here uses whole genome alignments and unrooted maximum likelihood trees which are then scanned for recombination. As such no reference is specified.

3. For the persistent mutations, what is their distribution across the species? That is, are the polymorphisms acquired during infection present in other sequenced isolates?

Authors response: The authors thank the reviewer for this suggestion and have added this information to the manuscript in the methods {lines: 447-452} results {lines: 174-180} and discussion {lines: 319, 320} sections. In the long-term study, there were persistent, non-synonymous mutations found in *rssA*, *nuoN*, *CRISPR-cas_1*, a large hypothetical protein (l-hp) and two hypothetical proteins. The non-synonymous mutation found in *rssA* has been seen in another *N. lactamica* gene found via BLASTp against the nr database. The specific location and non-synonymous, amino acid changes relative to that position of mutants of all other genes mentioned above were not identified in either sequence database blast of the entire *Neisseria* catalogue of pubMLST *Neisseria* or via non-specific BLASTp against the nr database. Therefore, these are most likely novel polymorphisms found during the course of this study.

4. The number in lines 140-147 do not agree with those in Fig. S3. Please correct or clarify.

(i) It appears as 97 versus 95 for strain Y92-1009, and 53 versus 32 for monoclonal colonization.

Authors response: We thank the reviewer for spotting this and have corrected the text accordingly.

5. Please provide more detail on the isolation of strains from the swabs. For instance:

(i) on average, how many colonies were present on the plated oropharyngeal swabs?

(ii) how was *N. lactamica* and *N. meningitidis* separated from other *Neisseria* species? Was it visual or was any additional molecular testing performed on the colonies?

(i) Authors response: We did not record this but we generally found in the range 1-20 colonies per plate.

(ii) The two bacterial species were initially isolated through culture on GC selective agar plates, which are inhibitory to the growth of the majority of other *Neisseria* species. Subsequently, colonies of *N. lactamica* were discriminated from all other *Neisseria* species on the basis of blue/white colony

formation after exposure to 5-bromo 4-chloro 3-indolyl B-D-galactopyranoside (X-Gal). Following exposure to X-Gal, *N. lactamica* colonies turn blue, whilst all other species of *Neisseria* remain white/pale grey. Speciation of putative meningococcal isolates was performed using biochemical testing (API strips, Biomerieux). Strain level specificity was achieved using a Y92-1009-specific PCR as described in Deasy et al (reference 17).

6. Line 266-273: Does this imply independent transfer events into the *N. lactamica* strain? One set captures in the initial collection and one set later? Please expand.

Authors response: This is a possibility and we have included a sentence in the text (lines: 291-292) to hypothesise this might reflect more than one transfer event and thus on-going allelic exchange. However, we feel it is not possible to go further with this given the data is derived from a single isolate at each sampling point.

8. The sequence similarity analysis suggests that L-hp is shared across strains in this species. The conclusion drawn that it is a HGT from *H. influenzae* requires more data. At this point the analysis simply suggests these are similar sequences, and these could be transferred across these species. At a minimum, does the GC content provide additional support for *H. influenzae* as a donor? Compare GC across gene to *Haemophilus*, difference against *avergae* Y92 GC

Authors response: We appreciate the speculative nature of this suggestion; however, we are keen to point out that HGT between the genera is not without basis and have cited accordingly. The reviewer's suggestion is useful however and we found the GC content (31.3%) to be skewed negatively compared to the overall GC content of the genome (52.2%) and closer towards the mean GC content for *H. influenzae* (38%). Interestingly, and as mentioned in the text there is precedence for this ancestral, interspecific transfer between *N. lactamica* and *H. influenzae*. We have therefore included the answer to this query in the manuscript (lines 332-338).

9. 291-302: This paragraph implies neutral and adaptive mutations for *rssA* and other persistent mutations. Please clarify language.

Authors response: The reviewer is right to highlight this, we are postulating adaptive (non-neutral) selection at these loci and have altered the wording in reflection.

10. Lines 221-225. Sentence is confusing to read. Please write more clearly specifying co-carried.

Author response: We have altered the wording such that it now reads "Two volunteers (36 and 291) were found to co-carry naturally acquired *N. lactamica* (ST-4192 and 11524, n = 9 isolates) in addition to commensal *N. meningitidis* (ST-41 and ST-10457, n = 6 isolates) (Supplemental Table 1)." We hope that this improves the readability.

11. Add a period to the end of the sentences in Discussion.

Author response: Done.

REVIEWERS' COMMENTS:

Reviewer #1 (Remarks to the Author):

The manuscript is much improved but the only major conclusion I glean from this work is that phase variation rates 'might' be higher during colonization than during in vitro growth. The rest of the data are interesting and should be reported but do not lead to important insights into the mechanisms allowing colonization or growth of this organism in the human host, its relationship to other colonizing organisms, or genetics.

Reviewer #2 (Remarks to the Author):

This study stands out as there are very few studies focused on bacteria that colonize the upper respiratory track, which capture and quantify gene transfer during human infections. Studies of evolution in human carriage are critical for vaccination programs.

This revised manuscript successfully addressed the comments and suggestions. The findings outline the extent and mechanism of gene exchange in a human carriage study. Further, they provide an important framework to interpret additional studies on *Neisseria* evolution likely to be performed in vitro or in model systems.